# Cucumber Fruit Size and Shape Variations Explored from the Aspects of Morphology, Histology, and Endogenous Hormones

**DOI:** 10.3390/plants9060772

**Published:** 2020-06-19

**Authors:** Xiaoxiao Liu, Yupeng Pan, Ce Liu, Yuanyuan Ding, Xiao Wang, Zhihui Cheng, Huanwen Meng

**Affiliations:** College of Horticulture, Northwest A&F University, Yangling 712100, Shaanxi, China; liuxiaoxiao@nwafu.edu.cn (X.L.); yupeng.pan@nwafu.edu.cn (Y.P.); liuce@nwafu.edu.cn (C.L.); dingyuanyuan@nwafu.edu.cn (Y.D.); wangxa@nwafu.edu.cn (X.W.)

**Keywords:** fruit development, fruit size, cell number, cell size, endogenous hormone

## Abstract

Fruit size and shape are important qualities and yield traits in cucumber (*Cucumis sativus* L.), but the factors that influence fruit size and shape remain to be explored. In this study, we investigated the dynamic changes of fruit size and shape from the aspects of morphology, cellular levels and endogenous hormones for nine typical cucumber inbred lines. The results show that fruit length had a strong positive correlation to the cell number in the longitudinal section of fruit throughout the four stages of 0, 6, 12, and 30 DAA (days after anthesis). However, the significant negative correlations were found between fruit length and the fruit cell size at 12 and 30 DAA. Furthermore, fruit diameter was positively correlated to the cell number in the cross section at all the investigated fruit growth stages. The indole-3-acetic acid (IAA) content showed significant positive correlations to the fruit length at all fruit growth stages of −6, −3, 0, 3, 6, 9 and 12 DAA, but IAA content and fruit diameter showed significant negative correlations for all the stages except for at −6 DAA. The trans-zeatin riboside (tZR), zeatin (ZT), gibberellic acid (GA_3_) and jasmonic acid (JA) content had a positive or negative correlation with fruit length or diameter only at certain stages. Neither fruit length nor diameter had significant correlations to abscisic acid (ABA) content. These results indicate that variations in fruit size and shape of different cucumber inbred lines mainly result from the differences in fruit cell number and endogenous IAA content. The present work is the first to propose cucumber fruit size and shape changes from the combined aspects of morphology, cellular levels, and endogenous hormones.

## 1. Introduction

Cucumbers (*Cucumis sativus* L.), harvested immaturely and consumed fresh or processed as pickles, is an important vegetable crop cultivated worldwide [1]. Fruit size and shape of cucumbers are usually determined by fruit length (FL), diameter (FD), or the length-by-diameter ratio (Fruit shape index, FSI) [2,3]. Fruit size is a crucial domestication trait in cucumber [4], and it displays tremendous variation, from 5 to 60 cm in length [5,6]. In addition, cucumber fruits also have marvelously diverse shapes, which vary from slightly flat, ellipsoid, obovoid, and round to long or extremely long [7]. The fruit size and shape are essential yield and quality traits for breeding and can also be classifying features to define different market classes of cucumbers. Each cucumber market type has its unique commercial standards in fruit length and diameter. For example, the American pickling cucumbers have relatively short and blocky fruits with FSI of 2.8 to 3.0; the fruits of the north China fresh market type cucumbers are long and slim with FSI > 6.0. Most Beit Alpha type (mini) cucumbers are harvested when the diameter is between 2 and 2.5 cm, and the length is between 13 and 15 cm [1,8]. The remarkable diversity of fruit size and shape made cucumber a sound biology system to reveal the genetic basis and regulating mechanism of fruit size and shape development.

Fruit development is classically characterized as proceeding through four stages: fruit set, which involves ovary development and fertilization; fruit growth due mostly to cell division that involves seed formation and early embryo development; fruit growth that is achieved predominantly by cell expansion; and the final maturation and ripening [9,10]. Cucumber fruit development starts from an enlarged inferior ovary, and also largely follows this canonical progression [2,7]. The fruit cell number and size jointly determine the final fruit size and shape, in which the rapid cell division phase provides cells for the next stage where the fruit growth occurs mainly by cell enlargement [11]. However, it is inconclusive whether the differences in fruit size and shape among various varieties are mainly due to the cell number or cell size, or both. Some studies showed that the differences in fruit size and shape were mainly determined by the cell number, such as melon [12], sweet cherry [13], peach [14] and pear [15]. In cucumber, some mutants with small-sized fruits were caused by an overall smaller cell size [16,17], while the small-sized fruit mutant with larger fruit cell size but lower cell number is also existed [18]. In addition, both the number and size of fruit cells were correlated to the final fruit size, which was also revealed in apple [19] and pineapple [20]. Corresponding to the cell number and cell size, respectively, the biological processes of cell division and cell expansion play important roles in the final formation of fruit size and shape [3,21]. In cucumber, the cell division process occurs most rapidly during the first several days but with minimal increase in fruit size, while the following exponential increase in fruit length coincided with the period of the highest increase in cell size [22,23,24]. Although recent studies have been targeted at observing histological regulation for cucumber fruit development, most of them focus on a unique mutant or few varieties, and no conclusive answers have yet been obtained to explain the cucumber fruit size and shape development and formation from the cellular levels.

Plant hormones play critical roles in every aspect of plant growth, development, and physiology, in which every cell is the product of the hormonal activity, and all living cells are responsive to hormones [25]. Auxin is involved in the regulation of plant growth and development, influencing cell expansion and division, cell elongation and differentiation, and a variety of physiological responses, thus significantly affecting the final shape and function of cells and tissues in all higher plants [25,26,27,28]. In tomato and Arabidopsis, auxin accumulation dramatically increases in the fertilized ovules, which stimulates auxin signaling and subsequent cell division and expansion to promote fruit growth [29,30]. Cytokinin is one of the so-called classic plant growth phytohormones and functions to promote the cell division and cell differentiation that are involved in almost all aspects of plant growth and development. Nowadays, multiple natural cytokinin species, including trans-zeatin (tZ), isopentenyladenine (iP), cis-zeatin (cZ), trans-zeatin riboside (tZR) and their conjugates have been identified [31]. Matsuo et al. [32] analyzed tZ, tZR, iP, isopentenyladenosine (iPR), dihydrozeatin riboside (DHZR), dihydrozeatin (DZ) concentrations in tomato, providing evidence that cytokinins are involved in cell division during fruit development. Studies also found that gibberellin (GA), abscisic acid (ABA), and jasmonic acid (JA) might also have important functions in controlling fruit growth and development [16,25,33,34,35]. However, there is currently very little information about endogenous hormones in regulating fruit size and shape development in cucumber. 

Therefore, the primary purpose of this study was to investigate the dynamic fruit size and shape changes in cucumber from aspects of morphology, cellular levels, and endogenous hormones. Nine typical cucumber inbred lines, including different market types and landraces, were selected and used in this study. We measured and calculated the fruit length (FL), diameter (FD), and the fruit length-by-diameter ratio (fruit shape index, FSI) at different fruit development stages. The endogenous hormone content of indole-3-acetic acid (IAA), tZR, zeatin (ZT), GA_3_, ABA, and JA for the ovaries or fruits bearing at different growth stages were also tested. Paraffin sections were conducted for these nine cucumber varieties to investigate the fruit cell number, size, and density on the transections and longitudinal sections of cucumber ovaries or fruits. The results of this study illustrate that the fruit size and shape variations of various cucumber inbred lines mainly result from the differences in fruit cell number and endogenous IAA content, and the fruit cell size and other investigated plant hormones also have related effects.

## 2. Results

### 2.1. Dynamic Changes in Fruit Size and Shape during the Process of Cucumber Fruit Development

Nine typical cucumber inbred lines were selected and used to observe the fruit size and shape changes among different fruit development stages. Although these cucumbers exhibited diverse fruit size and shape (Figure 1a), the growths of their FL and FD showed similar sigmoid growth curves, initiating from −6 to 3 DAA, exponentially increasing at 3 to 12 DAA, and continuously growing with a lower rate from 12 to 30 DAA. However, the incremental rate of FD decreased relatively slower than that of FL after 12 DAA (Figure 1b,c). Moreover, the dynamic changes in seed cavity diameter (SCD) from 0 to 30 DAA were similar to that of FD (Figure 1e). However, when compared with FL, FD, and SCD, the fruit shape index (FSI) exhibited a different growing curve, in which the highest point was reached at 6 to 12 DAA. For the FSI, both varieties with more extended (e.g., 9930, WI7123, NZ1, and WI7237) and medium-length fruits (WI7204) exhibited changes of rising first and then falling (Figure 1d), which indicates that the fruit growth ratio in the longitudinal direction was higher and lower than that in the transverse direction at the early and late developmental stages, respectively. In the case of shorter and round fruit inbred lines, including True Lemon, WI7120, WI7200, and Gy14, the FL and FD had a similar growth rate that resulted in relatively steady changes in FSI throughout the whole growing period.

Among these nine varieties, True Lemon, WI7120, and WI7200 showed little changes in FL after 12 DAA, while the other six inbred lines had continued growth, especially for the FL of WI7237, which still increased with a relatively higher rate (Figure 1b). On the other hand, all these nine inbred lines continued to increase for the FD until the 30 DAA, except for WI7200, that had a slower FD growth after the 12 DAA (Figure 1c). In general, the varieties with slender fruits such as WI7237, NZ1, WI7123, 9930, and WI7204 had larger final FL while having a smaller FD. Meanwhile, the inbred lines bearing fatter fruits, including Gy14, WI7200, WI7120 and True Lemon, had gone exactly the opposite way, presenting wider FD but shorter FL (Figure 1a–c). These nine inbred lines were also sorted based on their final FL, FD, FSI, and SCD. Moreover, the detailed ranking orders were listed in Appendix A, in which WI7237 had the longest FL and the narrowest FD, while True Lemon presented inversely with the shortest FL but the widest diameter. In addition, for the ranking of these nine varieties, the FSI and SCD were similar to the FL and FD, respectively (Appendix A).

Correlation analysis was performed based on the phenotypic data of FL, FD, FSI, and SCD collected at different fruit development stages for these nine cucumber cultivars. The detailed Pearson correlation coefficients between each trait pair are listed in Table 1. The negative correlation between FL and FD was not significant at the early fruit development stages (−6 to 9 DAA), while this correlation became significant once the fruits reached 12 DAA and remained so until the final mature fruit stage (30 DAA). Moreover, both trait pairs of FL vs. FSI and FD vs. SCD had strong positive correlations throughout the whole fruit growth period (−6 to 30 DAA), which might suggest that the FL and SCD play essential roles in the determination of FSI and FD, respectively. However, both FD vs. FSI and SCD vs. FSI showed significant negative correlations among the whole fruit developmental stages, which indicated that FD and SCD performed similar effects on fruit shape formation in cucumber.

### 2.2. Fruit Size and Shape Difference of Nine Typical Cucumber Varieties Revealed from the Histological Level

To clarify the histological changes in cucumber fruit growth, paraffin sections were conducted for the nine cucumber inbred lines. The cell number, size, and density were measured and calculated on the transections and longitudinal sections of cucumber ovaries or fruits. The fruit cell typical pictures, numbers, sizes, and densities at different development stages of these nine varieties are shown in Figure 2. For the cell number, there were relatively rapid increases from 0 to 6 DAA, while small changes were observed from 6 to 30 DAA except for the variety of WI7237, that continuously increased (Figure 2b). In the case of cell size and density, they kept increasing and decreasing among the whole fruit growing period, respectively (Figure 2c,d). The rapidly increasing stages of FL (3 to 12 DAA) involved both the cell number increasing stage and the following exponential increasing stage of cell size. This indicated that fruit cell numbers and sizes play important roles in the early and late fruit development stages, respectively, to determine the FL in cucumber. Furthermore, the final fruit cell size at the mature stage (30 DAA) was increased about 10–25 times compared to that at the ovary stage (0 DAA).

The phenotypic data of fruit cell number, size, and density at the ovary (0 DAA) and mature fruit (30 DAA) stages collected from those nine cucumber varieties are listed in the Table 2. Among these nine cucumber cultivars, the extremely long cucumber WI7237 had the most abundant fruit cell number in the longitudinal section and the highest fruit cell density in both cross and longitudinal sections; however, the fruit cell size of WI7237 was the smallest in both vertical and horizontal sections; the cucumber landrace True Lemon showed the smallest and largest fruit cell number and cell size in the longitudinal section, respectively; the North China fresh market type cucumber WI7123 had the lowest fruit cell density in the longitudinal section and the smallest fruit cell number in the cross-section, while the largest fruit cell number in the cross-section was shown in the US pickling cucumber Gy14; in addition, the Beit Alpha cucumber WI7204 showed the largest fruit cell size and the lowest cell density in the cross-section, respectively (Table 2). Overall, the cucumber varieties with different fruit sizes and shapes had different cell number, size, and density in both longitudinal section and cross-section of cucumber fruits, which indicated that the complex process of cucumber fruit size and shape determination in the cellular levels. Nevertheless, the fruit cell number, size, and density should be three primary factors in the final form of cucumber fruit size and shape.

Correlations between FL and fruit cell number, size, and density in the longitudinal section and between FD and fruit cell number, size, and density in the cross-section were calculated and are listed in Table 3. FL had a strong positive correlation to the fruit cell number in the longitudinal section throughout the four stages of 0, 6, 12, and 30 DAA. However, FL only showed significant correlations to fruit cell size, and density in the longitudinal section at the stages of 12 and 30 DAA, and the relations were negative and positive correlations, respectively. For the cross-section, FD had a positive correlation to fruit cell number at both stages of 0 and 30 DAA, while FD only showed positive and negative correlations to the fruit cell size at the 0 DAA stage and the fruit cell density at the 30 DAA stage, respectively. Overall, fruit cell number had correlations to FL and FD at all the fruit development stages, while the correlations of fruit cell size and density with FL and FD had a growth stage-dependent manner.

### 2.3. Cucumber Fruit Size and Shape Differences Explored from the Analysis of Endogenous Hormones

The dynamic changes in the levels of the endogenous hormones IAA, tZR, ZT, GA_3_, ABA, and JA in ovaries or fruits of nine cucumber inbred lines at different growth stages (−6, −3, 0, 3, 6, 9, and 12 DAA) are shown in Figure 3.

For the IAA content, although it showed a general trend of increasing first and then decreasing, there were some changes with the growth of cucumber fruits. The four varieties bearing longer fruits, WI7237, NZ1, WI7123, and 9930, had similar changes in IAA content, in which two content peaks emerged at the fruit initiating stage (−3 DAA) and rapidly growing stage (9 DAA), respectively. The IAA content gradually increased from 0 to 9 DAA, and then decreased rapidly after 9 DAA. WI7123, NZ1, and 9930 had the highest IAA increasing rate from 6 to 9 DAA, while WI7237 showed sharpest IAA rising from 0 to 6 DAA. The IAA content changes of cultivars with moderately long fruits, including Gy14 and WI7204, were relatively stable in the process of fruit growth, which only peaked at −3 DAA and then showed smoother changes since after DAA. For the shorter and rounder fruited inbred lines, WI7120 also had two peaks of IAA content which respectively showed up at 0 and 6 DAA, and the highest increasing rate occurred from 3 to 6 DAA, while the other two varieties, WI7200 and True Lemon, only showed one IAA content peak that was reached at 0 and 3 DAA, respectively; in addition, WI7200 had the fastest IAA increase from −3 to 0 DAA, and that of True Lemon occurred from 0 to 3 DAA. Compared with the longer fruit cultivars, the shorter ones had an earlier IAA content decrease. In general, at the later fruit growth stages (9 and 12 DAA), the relation between fruit size and IAA showed that the lengthier or larger fruit varieties had higher IAA content.

The tZR levels gradually decreased as the fruit developed from −6 to 12 DAA and kept almost unchanged since the 6 DAA. The four inbred lines of WI7200, WI7237, Gy14, and WI7120 had relatively higher tZR contents from −6 to 3 DAA. The ZT content of each variety showed a trend of decreasing first, being relatively stable, and then increasing again. Gy14 and NZ1 had a decreasing ZT content from −6 to 0 DAA and showed little changes from 0 to 6 DAA. The ZT content of the rest of the cultivars decreased from −6 to −3 DAA and then showed few changes from −3 to 6 DAA. All nine studied cucumber varieties showed a sudden increase in ZT content from 6 to 9 DAA. The ZT content of four varieties, Gy14, WI7204, WI7120, and True Lemon, continuously increased after 9 DAA, while the other five cultivars showed relatively steady changes in ZT content. 

For the GA_3_ content, all the varieties had a downtrend from −6 to 0 DAA, except for True Lemon and WI7200, which had an uptrend. From 0 to 9 DAA, only WI7120 showed a decreased GA_3_ content, and the remaining cultivars showed few changes. The GA_3_ content of True Lemon and WI7237 also remained little changed from 9 to 12 DAA, but the other inbred lines increased during this stage. The GA_3_ content of landrace WI7120 was much higher than that of other varieties at the stage of −6 to 3 DAA. Furthermore, the GA_3_ content of WI7237 remained lower than that of the remaining inbred lines throughout all the investigated fruit growth stages. The endogenous levels of ABA were highest at anthesis and then started to decrease soon after pollination, while 9930 is an exception, with its ABA content peaked before flowering (−3 DAA). Among all these nine cultivars, WI7120 and WI7200 had the highest and lowest ABA content at the day of anthesis, respectively. The JA levels behaved a general trend of increasing first and then decreasing. However, the fruit growth stages that showed the highest JA content were mostly different among the measured nine varieties. 

Correlations of FL and FD to the content of each endogenous hormone were also calculated and are listed in Table 4. The IAA content showed significant positive correlations to the FL at all seven fruit growth stages that were used for endogenous hormone evaluation. However, the IAA content and FD showed significant negative correlations at all the stages except for at −6 DAA. Whether the weak correlations between FL and FD and the content of tZR, ZT, and JA were significant or not or positive or negative, they were dependent mainly on the fruit growth stages. The content of GA_3_ had a negative correlation to FL from −6 to 3 DAA, and then the correlation was not significant anymore for the fruit growth stages from 6 to 12 DAA. Moreover, the significant negative correlation between GA_3_ and FD was only shown at the stage of −3 DAA. In addition, for all the involved fruit growth stages, neither FL nor FD had significant correlations to the content of ABA.

## 3. Discussion

Fruit size and shape, as an essential criterion to classify different market types, are important horticultural traits in cucumber. The notable diversity of fruit size and shape made cucumber a good biology system to reveal the genetic basis and regulating mechanism of fruit size and shape development. Fruit cell number, size, and density and endogenous hormones usually can directly influence the formation and development of final fruit sizes and shapes. However, the relations of cucumber fruit size and shape with fruit cell number, size, and density and endogenous hormones are still poorly understood in cucumber. In this study, we investigated the dynamic fruit size and shape changes in cucumber from aspects of morphology, cellular levels, and endogenous hormones.

Cucumber fruit develops from an enlarged inferior ovary, cell division occurs most rapidly before anthesis and then continues to increase to about 6 DAA and remained more or less constant thereafter [3,27,36]. After 6 DAA, the fruit cell size began to increase exponentially, which was consistent with the period of the fastest growth of cucumber fruit [3,22]. The pollinated cucumber fruits and parthenocarpic or non-pollinated fruits exhibited similar changes in length, diameter, cell number and cell size during cucumber fruit development [3,24,27]. The number of cells in each fruit did not differ significantly between the non-pollinated and pollinated groups. However, fruit cell size was larger in the pollinated group in winter season, but not significantly different from the non-pollinated group in summer season [27]. Here, we compared the morphological and cellular changes of fruits in nine typical cucumber inbred lines with various fruit sizes and shapes during the fruit development stages. The inbred lines with longer fruits showed higher cell number and cell division rate in the longitudinal direction than those with shorter fruits (Figure 1b and Figure 2b). WI7237, with the longest fruit, had the most abundant fruit cell number and the smallest fruit cell size in the longitudinal section, while True Lemon, with much shorter fruits, showed the smallest and largest fruit cell number and cell size in the longitudinal section, respectively. Furthermore, the positive correlations of FL and FD to fruit cell numbers at all the investigated fruit growth stages was highly significant (Table 3), which suggested that fruit cell number is strongly associated with the fruit size formation in cucumber. Overall, these results are consistent with the previous studies on melon [12], sweet cherry [13], peach [14] and pear [15]. 

Nevertheless, fruit cell size also plays a vital role in fruit size formation. Yang et al. [24] analyzed the fruit morphological and cellular changes of four different cucumber genotypes and found that both fruit cell number and size strongly correlated with the final formation of fruit size and shape. Similar results can also be revealed from the histological analysis of fruit size in pineapple [20]. In this study, we also observed that although the FL of WI7123 is longer than that of 9930, WI7123 has a smaller fruit cell number and larger cell size in the longitudinal section than 9930 (Figure 1b and Figure 2b,c). This suggests that both fruit cell numbers and sizes have effects on cucumber fruit size and shape, but the fruit cell number might play a more critical role in this development process. 

At the early stages of fruit development, plant hormones are usually involved in the direct or indirect regulation of fruit cell division and expansion processes and then further influence the fruit growth and determine the final fruit size and shape [9,37]. Auxin plays a vital role in many biological processes of plants, including embryo, root, flower, and fruit development. In addition, auxin is believed to promote the continued fruit expansion and enlargement following the finish of fruit setting stage [9]. Several studies have demonstrated the essential roles of the auxin signaling pathway in the development of fleshy fruits, especially the determination of fruit cell division and expansion [37,38,39,40]. In apple, some studies found that exogenously applied auxin during the end of cell division/early cell expansion phase can increase fruit size, and one auxin response gene Auxin Response Factor (*ARF106*) maps to a size-related quantitative trait loci (QTL) [39]. IAA content had significant positive correlations to FL throughout the stages of −6 to 12 DAA (Table 4). Meanwhile, IAA content showed significant positive correlations to cell number in the longitudinal section at both 6 DAA (0.67, *p* < 0.001) and 12 DAA (0.75, *p* < 0.001). Studies in tomatoes confirmed that increasing the IAA content released signals and stimulated cell cycle changes to promote cell division, but the cell expansion rate changed little [41], which indicated that IAA could promote cucumber fruit cell division. On the other hand, we observed that the two IAA content peaks emerged at the cucumber fruit initiating stage (−3 DAA) and rapidly growing stage (9 DAA) in inbred lines with longer fruits, respectively. Some studies have shown that the peak of auxin concentration coincides with the growth rate of cell elongation [40]. The rapid increase period of IAA content was consistent with the stage of exponential growth of cucumber fruit cell size in this study, which indicates that IAA may promote cucumber fruit cell elongation. Studies have shown that *CsFUL1^A^* negatively regulates fruit elongation under a certain expression window in cucumber. *CsFUL1^A^* binds to the CArG box in the promoter region of *SUPERMAN* (*CsSUP*), a regulator of cell division and expansion, to repress its expression. Additionally, *CsFUL1^A^* inhibits the expression of auxin transporters *PIN-FORMED 1* and *7*, resulting in decreases in auxin accumulation in fruits [17]. These data indicate that endogenous auxin is one of the important factors in controlling cucumber fruit size and shape development, and IAA content affects the cucumber fruit size mainly through affecting cell division and cell elongation. 

Arnau et al. [42] reported that the concentration of cytokinin (CTK) DHZR, dihydrozeatin nucleotide (DHZMP) and isopentenyladenine nucleotide (iPMP) are pervasive during the setting and exponential growth of peach fruits. In addition, in tomato, the amounts of CTK ribosides (tZR, iPR, and DHZR) before anthesis are relatively higher and reach their highest levels at 0 DAA, and then decrease drastically after anthesis [32]. This might suggest that before fertilization, CTK ribosides are required for ovaries to trigger and maintain cell division in the fruit primordia until the ovaries reach the mature size. Here, we observed that tZR levels in cucumber showed a sharp fall from −6 to 6 DAA and then became steady, which suggests that tZR might mainly involve in the processes of ovary and fruit cell division. ZT content showed significant positive correlations to cell number in the longitudinal section at 0 DAA (0.44, *p* < 0.05) and 6 DAA (0.43, *p* < 0.05). ZT may also promote cucumber fruit cell division. Some studies have shown that exogenous treatment of the −3 DAA female flowers with thidiazuron (*N*-phenyl-*N*′-1,2,3-thiadiazol-5-yl urea, TDZ), an inhibitor of CKX (cytokinin oxidase), partially complemented the short fruit phenotype, and the treatment had a much stronger effect on the *sf2* mutant than on WT fruit. This is consistent with a role for *SF2* in facilitating cell proliferation through modulation of CK contents [43]. Intriguingly, the ZT concentration was also relatively higher before anthesis and showed a sharp increase in the exponential growth stage (6 to 12 DAA), especially the inbred lines with wider fruit diameters. This shows that ZT played an important role in the process of cell expansion, which may promote cells expand radially. 

Gibberellin and auxin may act in parallel or in a sequential way in fruit set and growth [44]. Sun et al. [45] observed that the relatively higher content of GA_3_ kept decreasing in the early development stages (−2 to 6 DAA) of cucumber fruits. Here, we found that most cucumber inbred lines had a decreasing GA_3_ content before anthesis, which was kept relatively steady up to 9 DAA, then had an increasing trend from 9 to 12 DAA, when the IAA levels started to decrease. This suggested that GA_3_ may promote the cell expansion of cucumber fruits, which was consistent with the study by Serrani et al. [46], who found that the GA_3_-induced fruit mesocarp had smaller cell number and larger cell size. The content of GA_3_ had a negative correlation to FL from −6 to 3 DAA and to cell number in the longitudinal section at 0 (−0.59, *p* < 0.01), 6 (−0.43, *p* < 0.05) and 12 DAA (−0.39, *p* < 0.05). GA_3_ may inhibit fruit elongation by hindering cell division. This may suggest that GA_3_ plays a critical role in the development of fruit size and shape in cucumber.

Vriezen et al. [47] found that free-ABA levels are higher in the tomato ovaries at anthesis and decline after fertilization. Furthermore, ABA may play a role in the floral development in sweet cherry [48]. In strawberry, the ABA levels in strawberry fruits are also relatively higher on flower opening day but decreased when GA and IAA levels start to increase and stayed at extremely low levels throughout the S2 to S7 stages (2–4 DAA to 17–19 DAA) [49]. In this study, we also observed that the levels of endogenous ABA were highest at anthesis and then started to decrease soon after the pollination for most cucumber cultivars. This might verify the conclusion proposed by Vriezen et al. [47], which is that the function of ABA in fruit set might be antagonistic to that of auxin and gibberellin in order to keep the ovary in a temporally protected and dormant state, either to protect the ovary tissue or to prevent fruit development before pollination occurs.

## 4. Materials and Methods 

### 4.1. Plant Materials and Growth Conditions

Nine typical cucumber inbred lines were selected and used to observe the dynamics of fruit size and shape development from three aspects including morphology, histology, and physiology. Among the 9 inbred lines, both 9930 and WI7123 are North China fresh market types; Gy14, NZ1, WI7204 are representatives of US pickling, European greenhouse, and Beit Alpha market groups, respectively; and True Lemon, WI7120, WI7200, and WI7237 are landraces with varying fruit size and shape from different geographic regions [7]. We are very grateful to Yiqun Weng at the University of Wisconsin–Madison for providing the seeds of these nine cucumber inbred lines used in this study. All these cucumber lines were grown in the greenhouse, located in the Horticulture Farm of Northwest A&F University (HF-NWAFU), during the spring season (from February to May) of the year 2019. Three repetitions were planted for these cucumbers, in which 24 individuals per repetition were grown for each inbred line. Cultivation management in the greenhouse was performed based on standard practices.

To promote the fruit setting and development, all female flowers were hand pollinated. In addition, for each plant, we only kept one fruit bearing at the 10th to 15th nodes of the main vine to reduce the nutrient competition from other fruits and promoting its full development.

### 4.2. Measurement of Fruit Size and Shape

In cucumber, fruit length (FL) and fruit diameter (FD) are often used to describe fruit elongation and radial growth, respectively, during fruit development. Fruit shape could be conveniently defined using the fruit shape index (FSI), which is the ratio of FL to FD. To observe the morphological dynamic changes of fruit size and shape, we measured the FL and FD and calculated the FSI. In addition, data of the seed cavity diameter (SCD) were also collected in this study. All the above fruit size- and shape-related traits were collected at the time points of −6, −3, 0, 3, 6, 9, 12, and 30 DAA. Because the SCD at −6 and −3 DAA was too small to measure accurately, the data were not collected. Moreover, at least five ovaries or fruits per repetition were measured for each time point.

### 4.3. Paraffin Section and Measurement of Cell Numbers and Sizes

To observe the histological changes in cucumber fruit development, paraffin sections were conducted for these 9 cucumber lines. For each inbred line, samples were collected at four time points including 0, 6, 12, and 30 DAA, which covered the ovary, rapid growth, commercial mature, and mature stages. For each sample point, three different ovaries or fruits were picked and used for paraffin section analysis with the following procedures. Firstly, a small cube about 1 cm^3^ in size was cut from the mesocarp of the middle part of each fruit and immediately fixed in FAA (70% ethanol: formaldehyde: acetic acid with a volume ratio of 90:5:5) for 24 h, dehydrated through a series concentration of ethanol (70%, 85%, 95%, 100%, 100%, each for 1 h, respectively), transparent in xylene for 2 h (replace with new xylene after 1 h), and embedded in paraffin. Then, the longitudinal and cross sections with 8 μm in thickness were cut with a rotary microtome (Leica RT2235) and the sections were fixed on microslides and stained with toluidine blue for 30 min. Lastly, the well stained sections were sealed with resins and coverslips for photographing with the microscope (Olympus BX51).

The sections of each sample were photographed with the direction from the outer to inner of mesocarp. These obtained images were used to measure the cell numbers and sizes, in which at least three random locations were selected for measuring.

The cell number (N) and cell size (S) in given images were calculated using the software of Fiji (National Institutes of Health, Bethesda, MD, USA, https://imagej.net/Fiji) [50]. Average cell size was calculated as the total surface area divided by the cell number under a fixed observation area.

Cell number was determined using the formula of a circle for cross sections, and rectangle for longitudinal sections. The area of the whole fruit cross-section and longitudinal section (S′) was calculated by using the equation S′ = Total cross-sectional area − Seed cavity area of cross section or S′ = Total longitudinal section area − Seed cavity area on longitudinal section. The cell number in the whole fruit cross-sections and longitudinal sections (N’) was calculated by using the equation N/S = N’/S′. Cell density was determined by calculating the cell number under a certain area (500 μm × 500 μm).

### 4.4. Measurement of Endogenous Hormones

To check the changes in endogenous hormones in cucumber fruits during the early development stages (from ovary stage to commercial mature stage), ovary or fruit samples were collected at seven time points that included −6, −3, 0, 3, 6, 9, and 12 DAA.

Whole ovaries or fruits at −6, −3, 0, and 3 DAA were directly used as the samples for endogenous hormones analysis, and for each biological sample, usually three to five fruits from different plants were pooled together. Meanwhile, for the fruits at 6, 9, and 12 DAA, samples were collected as the mixtures of three parts of a fruit (the top, middle, and bottom). Three biological replicates were collected for each inbred line. All the ovary or fruit samples, collected around 9 to 10 am in the morning, were immediately frozen in liquid nitrogen and stored at −80 °C fridge in the laboratory for further use.

Auxin (indole-3-acetic acid, IAA), zeatin (ZT), trans-zeatin riboside (tZR), gibberellic acid (GA_3_), abscisic acid (ABA) and jasmonic acid (JA) were extracted and purified following the procedures described in Pan et al. [51]. Briefly, fruit tissues weighing about 1–2 g were weighed and ground into powder with mortar and pestle in liquid nitrogen; the powdered tissues were transferred into 50 mL tubes filled with pre-cooled (−20 °C) extraction solvent (isopropanol:H_2_O: concentrated HCl with a volume ratio of 2:1:0.002), in which the mass–volume ratio of sample:extraction solvent is 1:10 (g/mL); passive overnight extraction were continued about 16 h under −20 °C conditions [52]; then, centrifugation (4 °C, 10,000 rpm, 15 min) was performed to remove the tissue residues and to collect the supernatant containing hormones; a volume of 1.5 fold of dichloromethane (about 30 mL) to supernatant were added and shaken for 1 h at 4 °C, and then centrifuged for 15 min at 10,000 rpm to separate hormones with the water phase; the organic phases were collected into a round bottom flask and evaporated with a rotary evaporator under vacuum at 38 °C [53]; the obtained dried residues were resolved in 1 mL methanol and purified using a 0.22 μm syringe filter. The above purified hormones were measured by HPLC-ESI-MS/MS (QTRAP 5500, AB Sciex, Boston, MA, USA) with the setting conditions listed in Appendix A.

### 4.5. Statistical Analysis

The analysis of variance (ANOVA) was performed in R program (The University of Auckland, Auckland, New Zealand) with the basic function of “*aov*”. The ‘*pairs.panels*’ function in ‘*psych*’ R package (Northwestern University, Evanston, IL, USA) was used for analyzing the correlations among different traits (https://cran.r-project.org/web/packages/psych/). The R package of ‘*ggplot2*’ was used to plot the figures of statistical analysis results [54].

## 5. Conclusions

In conclusion, our results indicate that the fruit size and shape variations of different inbred lines in cucumber mainly result from the fruit cell number and endogenous IAA content. However, undeniably, fruit cell size and other endogenous hormones, including tZR, ZT, GA_3_, ABA and JA, also have critical effects on the growth of cucumber fruits at different developmental stages. This study could provide more valuable information for further uncovering the molecular regulatory mechanisms of fruit development in cucumber.

## Figures and Tables

**Figure 1 plants-09-00772-f001:**
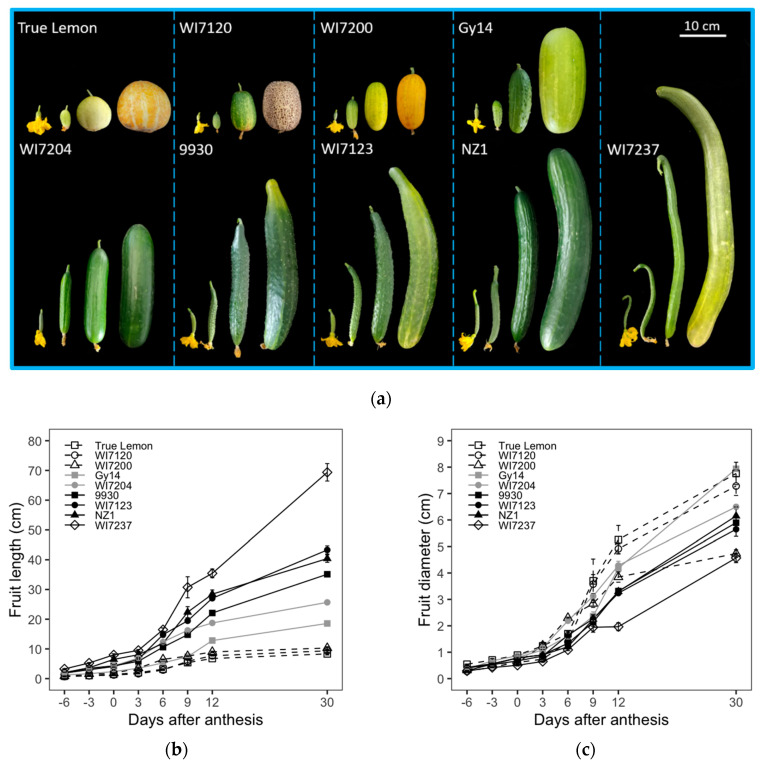
Dynamic changes of fruit size and shape related traits among nine typical cucumber inbred lines. (**a**) The ovary and fruit pictures of these 9 cucumber inbred lines, which were taken at 0, 6, 12 and 30 DAA, respectively. (**b**–**e**) The dynamic changes in fruit length (FL), diameter (FD), fruit shape index (FSI), and seed cavity diameter (SCD), respectively. Each value is the mean ± SE of 3 fruits.

**Figure 2 plants-09-00772-f002:**
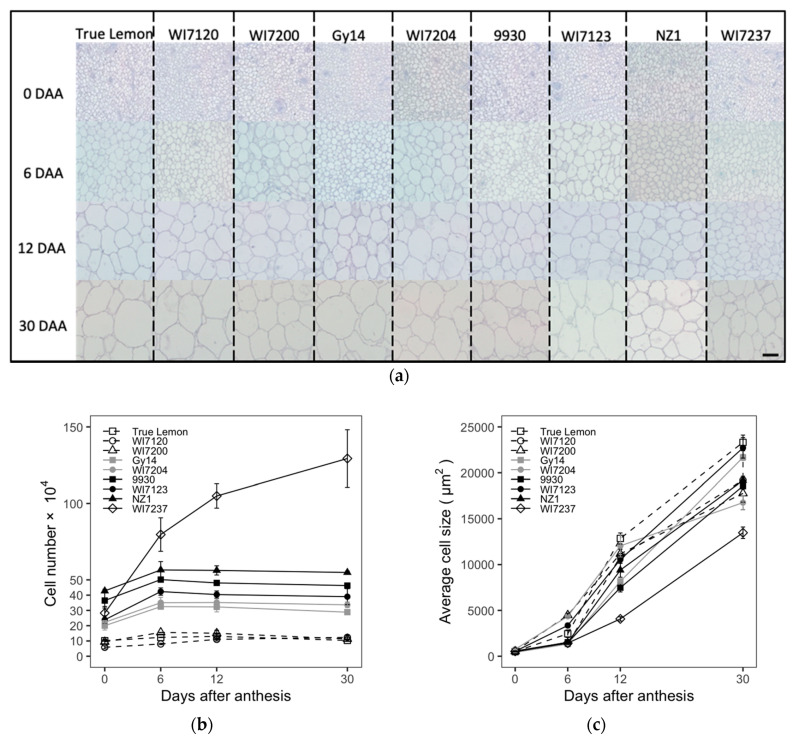
The dynamic cellular changes of fruit cells in the longitudinal sections among nine cucumber inbred lines. (**a**) The represented ovary and fruit cell pictures of different development stages. (**b**) Cell number. (**c**) Average cell size. (**d**) Cell density. Each value is the mean ± SE of 3 fruits. Bar = 100 μm.

**Figure 3 plants-09-00772-f003:**
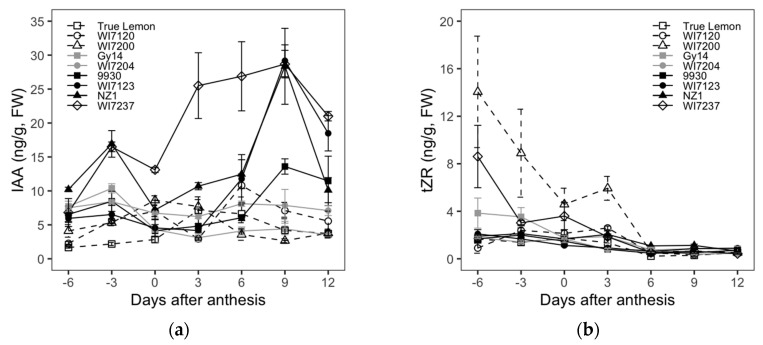
Comparative analysis of the endogenous hormone changes for the nine cucumber inbred lines at different fruit development stages. (**a**–**f**) Changes in the indole-3-acetic acid (IAA), trans-zeatin riboside (tZR), zeatin (ZT), gibberellic acid (GA_3_), abscisic acid (ABA) and jasmonic acid (JA) content. Each value is the mean ± SE of 3 fruits. FW, fresh weight.

**Table 1 plants-09-00772-t001:** Pearson correlation coefficients for fruit size- and shape-related traits at different development stages among nine typical cucumber inbred lines.

	−6 DAA	−3 DAA	0 DAA	3 DAA	6 DAA	9 DAA	12 DAA	30 DAA
FL × FD	ns	ns	ns	ns	ns	ns	−0.85 ***	−0.57 **
FL × FSI	0.95 ***	0.95 ***	0.93 ***	0.92 ***	0.90 ***	0.93 ***	0.92 ***	0.98 ***
FD × FSI	−0.49 **	−0.44 *	−0.50 **	−0.58 **	−0.57 **	−0.58 **	−0.88 ***	−0.65 ***
FL × SCD	-	-	−0.40 *	−0.54 **	−0.59 **	−0.53 **	−0.88 ***	−0.67 ***
FD × SCD	-	-	0.88 ***	0.86 ***	0.86 ***	0.94 ***	0.92 ***	0.93 ***
FSI × SCD	-	-	−0.55 **	−0.69 ***	−0.82 ***	−0.68 ***	−0.87 ***	−0.75 ***

FL, fruit length; FD, fruit diameter; FSI, fruit shape index; SCD, seed cavity diameter; -, data not recorded and available; ns, *, **, *** R value not significant, or significant at *p* < 0.05, *p* < 0.01, *p* < 0.001, respectively.

**Table 2 plants-09-00772-t002:** Cell number, average cell size and cell density of longitudinal and cross sections for nine cucumbers at anthesis (0 DAA) and mature stage (30 DAA).

	Longitudinal Section						Cross Section					
	CN		CS (μm^2^)		CD		CNc		CSc (μm^2^)		CDc	
	0 DAA	30 DAA	0 DAA	30 DAA	0 DAA	30 DAA	0 DAA	30 DAA	0 DAA	30 DAA	0 DAA	30 DAA
True Lemon	101,771 ± 3568de	103,756 ± 5600c	516 ± 29bc	23,313 ± 793a	453 ± 19ab	9 ± 0bc	61,392 ± 11,724a	77,557 ± 11,498abc	808 ± 78b	33,698 ± 1793a	333 ± 26bc	5 ± 1bc
WI7120	58,380 ± 10,094e	123,848 ± 12,756c	501 ± 18bc	19,198 ± 3201abc	515 ± 7ab	11 ± 3bc	38,193 ± 5113abc	82,685 ± 7605ab	470 ± 33c	31,194 ± 5507a	515 ± 5a	7 ± 1bc
WI7200	94,182 ± 17,256de	113,208 ± 372c	537 ± 25bc	17,774 ± 963abc	514 ± 19ab	10 ± 1bc	26,782 ± 2086bc	39,074 ± 3228d	695 ± 47bc	28,449 ± 3077a	366 ± 17bc	7 ± 1bc
Gy14	199,937 ± 28,513cd	288,472 ± 3758bc	420 ± 43c	21,673 ± 334ab	587 ± 47a	8 ± 1bc	42,940 ± 3305abc	97,720 ± 4891a	612 ± 21bc	32,874 ± 2094a	426 ± 12ab	7 ± 1bc
WI7204	222,032 ± 12,152cd	336,646 ± 19,155bc	755 ± 14a	16,764 ± 777bc	389 ± 33b	12 ± 2bc	40,806 ± 186abc	47,858 ± 3972cd	1154 ± 108a	38,785 ± 1655a	223 ± 10d	4 ± 1c
9930	364,702 ± 49,231ab	462,671 ± 12,044b	483 ± 24bc	18,559 ± 301abc	466 ± 49ab	10 ± 1bc	50,417 ± 6673ab	61,335 ± 8972bcd	756 ± 23bc	27,948 ± 3355a	323 ± 6c	8 ± 1ab
WI7123	238,546 ± 26,172bc	390,007 ± 52,163bc	540 ± 38bc	22,717 ± 1117ab	543 ± 47ab	7 ± 1c	31,285 ± 2635bc	38,584 ± 5036d	782 ± 44b	37,567 ± 1868a	348 ± 33bc	6 ± 1bc
NZ1	427,594 ± 10,155a	548,538 ± 8826b	575 ± 18b	19,164 ± 462abc	378 ± 13b	15 ± 1ab	35,014 ± 5845abc	56,450 ± 6056bcd	855 ± 81b	31,043 ± 1865a	306 ± 25cd	5 ± 0bc
WI7237	282,883 ± 42,592bc	1,293,214 ± 187,982a	536 ± 19bc	13,474 ± 626c	562 ± 32a	21 ± 2a	20,686 ± 1213c	52,316 ± 7330bcd	739 ± 28bc	25,685 ± 2479a	327 ± 19c	11 ± 1a

CN, CS and CD, cell number, average cell size and cell density of the longitudinal section, respectively; CNc, CSc and CDc, cell number, average cell size and cell density of the cross section, respectively; letters represent significant difference at *p* < 0.05, Tukey’s test.

**Table 3 plants-09-00772-t003:** Pearson correlation coefficients among fruit length and diameter and the cell number, size, and density at different development stages.

	0 DAA	6 DAA	12 DAA	30 DAA
FL × CN	0.77 ***	0.84 ***	0.91 ***	0.92 ***
FL × CS	ns	ns	−0.63 ***	−0.44 *
FL × CD	ns	ns	0.72 ***	0.62 ***
FD × CNc	0.65 ***	-	-	0.77 ***
FD × CSc	0.46 *	-	-	ns
FD × CDc	ns	-	-	−0.48 *

FL, fruit length; FD, fruit diameter; CN, CS and CD, cell number, average cell size and cell density of the longitudinal section, respectively; CNc, CSc and CDc, cell number, average cell size and cell density of the cross section, respectively; -, data not recorded and available; ns, *, **, *** R value not significant, or significant at *p* < 0.05, *p* < 0.01, *p* < 0.001, respectively.

**Table 4 plants-09-00772-t004:** Pearson correlation coefficients among fruit length and diameter and endogenous hormones at different development stages.

	−6 DAA	−3 DAA	0 DAA	3 DAA	6 DAA	9 DAA	12 DAA
FL × IAA	0.54 **	0.75 ***	0.53 **	0.59 **	0.60 ***	0.80 ***	0.83 ***
FL × tZR	ns	ns	ns	ns	0.43 *	ns	ns
FL × ZT	−0.44 *	ns	ns	ns	0.55 **	ns	ns
FL × GA_3_	−0.43 *	−0.38 *	−0.60 ***	−0.63 ***	ns	ns	ns
FL × ABA	ns	ns	ns	ns	ns	ns	ns
FL × JA	−0.64 ***	ns	ns	ns	0.48 *	ns	ns
FD × IAA	ns	−0.39 *	−0.59 **	−0.38 *	−0.54 **	−0.49 **	−0.75 ***
FD × tZR	ns	ns	−0.43 *	ns	ns	−0.56 **	ns
FD × ZT	−0.39 *	ns	0.41 *	ns	ns	−0.65 ***	ns
FD × GA_3_	ns	−0.43 *	ns	ns	ns	ns	ns
FD × ABA	ns	ns	ns	ns	ns	ns	ns
FD × JA	ns	ns	ns	ns	ns	−0.50 **	−0.49 **

FL, fruit length; FD, fruit diameter; IAA, indole-3-acetic acid; tZR, trans-zeatin riboside; ZT, zeatin; GA_3_, gibberellic acid; ABA, abscisic acid; JA, jasmonic acid; ns, *, **, *** R value not significant, or significant at *p* < 0.05, *p* < 0.01, *p* < 0.001, respectively.

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
