# Peer review of "Cucumber Fruit Size and Shape Variations Explored from the Aspects of Morphology, Histology, and Endogenous Hormones"

_plants, 2020, doi:10.3390/plants9060772_

Round 1

Reviewer 1 Report

In this manuscript, the authors performed a morphological, histological, and plant hormone analysis to characterize the events that occurred during the cucumber fruits development. From the results of these analyses, the authors revealed the following three major findings: (i) fruit cell number is strongly correlated with the fruit size formation in cucumber, (ii) two endogenous auxin peaks at -3 DAA and 9 DAA in inbred lines bearing longer fruits may promote a cucumber fruit cell elongation, and (iii) IAA content shows a positive correlation to the fruit length. Results presented in this study are clear and conclusions are reasonable, drawn from the results. I think this study is acceptable for publication in Plants, but I’d like the authors to consider the following things.

  1. In this study, the authors analyzed hand-pollinated fruits. However, many cucumber cultivars show a parthenocarpic phenotype. I want to know whether such parthenocarpic fruits exhibit the same morphological and histological features as in the hand-pollinated fruits. Please discuss this aspect by citing previous studies if possible.

  1. I also want to know how the exogenous treatment of plant hormones (especially auxin) to the fruits of nine cucumber inbred lines affect the fruit shape. If the exogenous auxin treatment changes a morphological change of the shorter and rounder fruits to the longer fruits, the amount of auxin determines the shape of the cucumber fruits and if not, other factors like auxin signaling pathway also plays an important role. I understand this kind of analysis is outside the author’s research objectives, so please discuss this aspect by citing previous studies if possible.

Minor points:

(Throughout the manuscript) The authors frequently used the term ‘cell area’, but the term ‘cell size’ is easier for readers to understand.

(Figure 1, 2, and 3) Symbol represents W17237 is hard to recognize. Please change the symbol.

(Figure 2a) “100 μm” overlaps with the figure.

(Table 1 and 2) What does the symbol ‘–’ mean? Please include the information in the footnote. Also, if this symbol means “analysis was not carried out in this stage”, the authors should mention such information in Materials and Methods.

(Conclusion section) Although the guidelines for Authors of Plants placed Conclusion after Materials and Methods, I think this format is not popular. As the Discussion section is not so long or complex, I suggest the authors that moving a paragraph of Conclusion to the last part of Discussion.

Reviewer 2 Report

Dear Authors,

Your manuscript describes a well-designed research. The logical structure of the Ms. is appropriate and the quality of data presentation is also good.

The introduction is focused, the methods are presented is sufficient detail. The tables, graphs and photograps are well-edited.

Overall, I found this study and the manuscript valuable. I have some minor comments:

-in the abstract results of hormones other than IAA should be mentioned

-please, emphasize the novelty of this study! What are the newer information compared to the previous literature?

I suggest to use phrases like "this study examined for the first time..."

-in conclusion section authors did not mention JA. 
